# Evaluation of a Group Therapy for Work-Related Mental Disorders

**DOI:** 10.3390/ijerph20032673

**Published:** 2023-02-02

**Authors:** Sinja Hondong, Eva Morawa, Silke Kastel-Hoffmann, Anja Kandler, Yesim Erim

**Affiliations:** Department of Psychosomatic Medicine and Psychotherapy, University Hospital of Erlangen, Friedrich-Alexander University Erlangen-Nürnberg (FAU), 91054 Erlangen, Germany

**Keywords:** anxiety, burnout, depression, mental health treatment, work-related mental disorders

## Abstract

**Objectives:** Work-related mental distress is one of the most dominant reasons for sick leave and early retirement. Specialized therapy programs for work-related mental health problems are rare, especially in a group setting. This study evaluates the severity of depression, anxiety, somatization and burnout symptoms before and after a work-related group therapy program. **Methods:** Patients of a psychosomatic outpatient clinic with work-related mental disorders completed 12 sessions of a manual-based group training with reference to the workplace. Data were collected using the Patient Health Questionnaire-9 (PHQ-9), Patient Health Questionnaire-15 (PHQ-15), General Anxiety Disorder Scale-7 (GAD-7) and the Maslach Burnout Inventory (MBI) before (T1) and directly after the intervention (T2). **Results:** Overall, 48 participants completed the intervention. The participants’ symptoms of depression (T1: M = 11.06, SD = 6.19, T2: M = 8.92, SD = 8.17; *p* < 0.001, *d* = 0.53) and anxiety (T1: M = 9.94, SD = 5.18, T2: M = 7.13, SD = 5.69; *p* = 0.001, *d* = 0.49) as well as their emotional exhaustion (T1: M = 4.63, SD = 0.95, T2: M = 4.05, SD = 1.35; *p* < 0.001, *d* = 0.55) decreased significantly, and the difference was clinically relevant at T2. For cynicism (T1: M = 3.93, SD = 0.99, T2: M = 3.70, SD = 1.32; *p* = 0.14, *d* = 0.22) and personal fulfillment at work (T1: M = 4.30, SD = 0.83, T2: M = 4.41, SD = 0.94; *p* = 0.24, *d* = 0.17), the difference between T1 and T2 was not significant. Women benefited more than men (PHQ-9: *p* < 0.001, *d* = 0.96; GAD-7: *p* < 0.001, *d* = 0.91; PHQ-15: *p* < 0.001, *d* = 0.76) from the training. **Conclusions:** Participants’ mental health symptoms were substantially reduced during the course of the work-related group therapy. As mental health problems account for the largest group of work disability days, the potential of group therapy should be better exploited in health care services.

## 1. Introduction

Mental diseases are becoming increasingly important in the work environment [1]. On the one hand, common mental disorders (CMDs) such as depression or anxiety disorders are the leading cause of disability in Europe, as can be seen in the increasing number of mental disorder sick leave and disability pensions [2]. On the other hand, there is growing evidence for workplace-related factors that are associated with the development, exacerbation or continuation of mental disorders [3]. Examples include high job demand, low job control, job insecurity, atypical working hours, bullying and role stress [4]. These work characteristics are also related to the burnout dimensions of emotional exhaustion, cynicism and personal fulfillment at work [5,6]. The eleventh version of the International Classification of Diseases (ICD-11) recognizes the importance of the work setting for mental disorders by listing burnout as a workplace phenomenon [7]. The previously mentioned burnout dimensions must be seen in the context of chronic work stress that has not been adequately managed.

Occupational mental health has predominantly been studied in the context of health care workers. In studies with this specific group, mindfulness-based and lifestyle interventions have been found to be helpful to reduce stress and enhance overall well-being in nurses or physicians [8,9,10]. These interventions rarely focused on the treatment of pronounced work-related mental disorders, and it is not clear whether the results can be transferred to other occupational groups. Two recent pilot studies have shown that work-related interpersonal group therapy can reduce symptoms of depressive disorders and improve workplace-related factors such as improved ability to work, self-efficacy regarding to work and sleep complaints [11,12]. There is evidence that a group intervention of four sessions for work-related anxiety can reduce sick leave if there is no other mental illness [13]. A comprehensive study is currently investigating the influence of psychotherapeutic consultation at work [14].

By a specialized intervention for work-related disorders, we define an intervention that specifically refers to the conditions at work, e.g., with a detailed work history and analysis of problems at work, with detailed discussion of the psychological meaning of work and finally preparing the return to work. There are currently few manualized and scientifically proven interventions for the treatment of occupational mental problems [15]. The aim of the intervention used in this study is to strengthen employees’ personal resources so that they can better cope with work-related stress or disorders [16]. Our main scientific question was to explore mental distress (depression, anxiety and somatization) and burnout symptom change in participants after a manualized work-related, psychotherapeutic group training. We assumed that after the intervention symptoms of depression, anxiety and somatization as well as work-related emotional exhaustion and cynicism will decrease, while personal fulfillment should increase.

Women experience more stress at work than men due to low support from superiors. Men are more likely to suffer higher life stress than women when job insecurity is high, while women experience more life stress when job strain is high [17,18,19]. Since those exemplary gender differences have already been shown in the context of workplace-related disorders, we examined gender-based differences for the change of the interested outcomes as additional analyses.

## 2. Method

### 2.1. Participants

Participants were recruited in the psychosomatic outpatient clinic where they were supposed to participate in a specialized group therapy. They were recruited by face-to-face interviews conducted before the beginning of therapy. All participants were at least 18 years old and met the criteria of a mental disorder with work-related factors. These are common mental disorders with an explicit reference to the workplace. This was the case if either the diagnoses Z56 (stressful arrangement of working hours, difficult working conditions and disagreements with superiors or work colleagues) or Z73 (problems related to difficulties in coping with life) could be diagnosed additionally to a common mental disorder [1]. All participants were employed, but some of them were on sick leave. Participants had to be excluded from the study if (a) there was an increased risk of suicide, or (b) there was comorbid psychotic disorder or additional acute substance dependence.

### 2.2. Data Collection

The data collection took place in the psychosomatic outpatient clinic of the University Hospital of Erlangen. The recruitment location is a city of 130,000 inhabitants in Bavaria in the south of Germany. Due to many large international companies and a large university and university hospital, the topic of workplace-related psychotherapeutic interventions is highly relevant locally.

Psychometric measurements were taken at the beginning of the study (T1) and at the last session (T2). The diagnosis was based on a semi-structured interview guide which is carried out in all preliminary interviews at the psychosomatic clinic. Interested persons were added to a waiting list and contacted by phone at the beginning of the group therapy.

Two weeks before the start of the group, the participants were sent questionnaires which they filled out and brought to the first group session. In the eleventh group session, the participants were given the same questionnaires for T2 and were asked to return them at the last meeting.

### 2.3. Intervention

The structure and content of the groups was based on the manual *mental stress in everyday working life—training manual for strengthening personal resources* [16]. The manual is designed as a cognitive behavioral therapy to promote personal resources that can be offered in a mixed individual and group setting or purely as group training.

Group therapy was led by licensed psychotherapists, psychologists (diploma or M.Sc.) in advanced training or physicians with psychotherapeutic skills. The group size varied between five and nine people. The group therapy was held weekly for twelve sessions of 90 min each. A total of 11 groups took place. To standardize and test the feasibility of the manual, a pilot group with the same content as all other groups was conducted. This group consisted of seven participants who were not included in the analyses.

The group therapy consisted of four modules. In module 1: *Work—Desire or Burden?* the individual meaning of work as well as basic needs and their meaning in the work context were explored. In module 2: *On the Trail of Stress*, psychoeducational knowledge about stress and a psychosomatic understanding of symptoms of functional body complaints were taught. In addition, individual personal stressors were analyzed. In module 3: *Coping with Stress*—*Building Up Resources*, stress management strategies such as problem-solving training, time management, communication training and relaxation techniques were taught and practiced. In module 4: *Recognizing Opportunities*, the handling of unsuccessful solution attempts and the proactive design of the work situation were discussed. Between sessions, participants were encouraged to do homework such as observation protocols or other worksheets.

### 2.4. Measures

#### 2.4.1. Diagnosis

Within the interviews, clinical diagnoses were assigned. During the interview, symptoms and medical history, eating behavior, psychosomatic history, organic self-history, substance and medication use, social situation, biographical aspects and psychopathological findings were assessed.

#### 2.4.2. Sociodemographic and Occupational Variables

All sociodemographic data were collected using a questionnaire comprising 10 items. These included information on age, gender, living situation, and marital status as well as the number of children. Migration status was assessed based on the Microcensus of the Federal Statistical Office [20] and included migrants, children of migrants and persons born in Germany as foreigners. Occupational questions concerned school-leaving qualification, vocational qualification, current occupation and current employment status. The question about the current occupational group was divided into seven categories ranging from unskilled workers to employees with extensive management and decision-making authority. All participants assigned themselves to a category. The categories were specified with respective examples.

#### 2.4.3. Depression, Anxiety and Somatization

To record symptoms of depression, somatization disorder and anxiety disorder, the respective instruments of the Patient Health Questionnaire were used (PHQ-9 [21]; PHQ-15 [22]; GAD-7 [23]). The PHQ-9 (9 items) measuring depressive symptoms and the GAD-7 (7 items) measuring anxiety symptoms can be rated on a four-point frequency scale from 0 (“not at all”) to 3 (“almost every day”). The PHQ-15 (15 items) for measuring somatization symptoms can be answered on a three-point scale from 0 (“not impaired”) to 2 (“severely impaired”). As to the original validation studies, a value greater than or equal to 10 represents the cut-off value for clinically relevant symptom expression in all three questionnaires.

#### 2.4.4. Burnout Dimensions

With the MBI-General Survey (MBI-GS-D; original English version see [24], German version see [25]), interdisciplinary investigations of the burnout syndrome can be conducted. It includes 16 items that can be assessed on six-level frequency scales with poles 1 (“never”) and 6 (“very often”). The items can be summarized on three scales (emotional exhaustion, cynicism and personal fulfillment). Values above 3.5 on the scales of emotional exhaustion and cynicism are considered as an indication of burnout, while values above 3.5 on the scale of personal fulfillment indicate that burnout is not present. This value was given by the author on direct enquiry [26].

### 2.5. Statistical Analysis

The statistical software IBM SPSS Statistics V. 28 (IBM, Armonk, United States) was used to analyze the data. At T1, there were 2.1%–6.3% missing values in PHQ-15 items, no missing values were observed in PHQ-9 and GAD-7. In the MBI, 2.1% of three items were not answered. At T2, there were 2.1%–8.3% missing values in PHQ-15, 2.1%–4.2% missing values in PHQ-9 and 2.1% were missing for two items in GAD-7 and in the MBI. The missing values were imputed using the expectation-maximization algorithm. Absolute numbers and percentages, means and standard deviations were calculated to describe sociodemographic data and the sample. McNemar tests were performed to test the significance of a change in symptom manifestation for dichotomized variables. T-tests for dependent samples were used to calculate differences between measurement time points for continuous variables. Cohen`s d effect sizes were determined to measure the effect sizes (*d* ≥ 0.2 = small, *d* ≥ 0.5 = medium, and *d* ≥ 0.8 = large effect size) [27]. The calculations were based on a significance level of *p* < 0.05 (two-tailed). In order to adapt the PHQ-15 questionnaire in a gender-neutral way, we additionally calculated a sum score without the item “menstrual cramps”. Since the results did not differ significantly and the version with 15 items is internationally recognized, these additional analyses are only shown in the tables. In addition to the analyses of the participants, intention-to-treat analyses and group comparisons for the descriptive variables were also calculated in order to exclude selection bias. The results can be found in Appendix A.

## 3. Results

### 3.1. Response Rate

A total of 78 individuals underwent the intervention. Of these, seven were in the pilot group and are not included in the analyses. Of the remaining 71 persons, the sample of completers included 48 participants (completed both measurement points). The response rate was 67.6% accordingly.

### 3.2. Sociodemographic and Occupational Variables

The sample consisted of 47.9% women and 52.1% men (Table 1). The participants had a mean age of M = 48.27 years (SD = 8.45, range: 26–59). Almost half were married (47.9%), while smaller proportions were single (22.9%) or divorced (22.9%). Nearly two-thirds had children (62.5%), most of whom had one (43.3%) or two children (40%). Five individuals reported a migration background (two from Romania and one person from Italy, Poland, and Turkey). The majority of the sample graduated middle school (39.6%) or secondary school (54.2%). Over one-third of the participants had completed an apprenticeship (37.5%) or a university degree (39.6%); 14.6% had a master’s degree. Only one person had no professional qualification. Accordingly, a part of the participants worked as employees with qualified activities (50.0%) or as employees with highly qualified activities with a management function (29.2%). One-tenth were even employed with extensive management activity and decision-making authority (10.4%). Three quarters of the sample worked full time (>35 h), and one quarter worked part time (15–34 h).

### 3.3. Diagnoses

The main diagnosis was an affective disorder (62.6%) (Table 2). In addition, 20.8% had an anxiety disorder as the main diagnosis and 8.3% had a diagnosis from the somatoform field. Two-thirds (66.7%) had one diagnosis, 25.0% had two diagnoses and 8.3% had three diagnoses.

### 3.4. Mental and Work-Related Distress at T1

At baseline, the mental distress of the sample was moderate to moderately severe in terms of depression, anxiety and somatization (PHQ-9: M = 11.06, SD = 6.19; GAD-7: M = 9.94, SD = 5.18, PHQ-15: M = 10.81, SD = 5.61) (Table 3). Burnout scores were high on the emotional exhaustion subscale (M = 4.63, SD = 0.95). Mean scores of cynicism were in the middle range (M = 3.93, SD = 0.99). At the same time, personal fulfillment was in the upper-middle range (M = 4.30, SD = 0.83).

### 3.5. Mental and Occupational Distress at T2

At the second measurement point, the mean values for depression, anxiety and somatization were lower than at T1 (PHQ-9: M = 8.29, SD = 6.27; GAD-7: M = 7.13, SD = 5.69, PHQ-15: M = 9.85, SD = 6.19). The mean scores for emotional exhaustion (M = 4.05, SD = 1.35) and cynicism (M = 3.70, 1.32) were also lower at T2. Personal fulfillment was almost unchanged (M = 4.41, SD = 0.94).

### 3.6. Changes of Mental Distress Measured in Prevalence

The frequency of depressive symptoms above the cut-off-value showed a reduction from 60.4% to 37.6% from T1 to T2 (*p* = 0.007) (Figure 1). Pathological anxiety was reduced from 58.3% above the cut-off at T1 to 33.4% at T2 (*p* = 0.012). More than half of the participants (52.1%) reported symptoms of somatization above the cut-off at T1, which did not significantly change at T2 (50.0%, *p* = 0.58).

Concerning burnout symptoms, the proportion of individuals with high emotional exhaustion was decreased from 83.3% at T1 to 64.6% at T2 (*p* = 0.004). The frequency of clinically relevant cynicism was 72.9% at T1 and 54.2% at T2 (*p* = 0.012). The frequency of personal fulfillment at work did not significantly change from T1 to T2 (89.6% to 87.5%, *p* = 1.00).

### 3.7. Changes of Mental Distress Measured in Symptom Manifestation

Depression levels decreased significantly and clinically relevant from the first to the second measurement time point (*p* < 0.001, *d* = 0.53) (Table 4). Anxiety symptomatology also decreased substantially over the course of training (*p* = 0.001, *d* = 0.49). Somatization levels were higher before training than after training for the entire sample (*p* = 0.12, *d* = 0.23).

Emotional exhaustion decreased significantly and clinically relevantly over the course of the intervention (*p* < 0.001, *d* = 0.55). Scores on the cynicism subscale decreased only slightly from the first measurement time point to the second measurement time point (*p* = 0.14, *d* = 0.22). Personal fulfillment also increased slightly after the intervention (*p* = 0.24, *d* = 0.17).

### 3.8. Gender Differences

Symptom levels improved meaningfully for women on all scales (somatization: *p* = 0.002, *d* = 0.76; depression: *p* < 0.001, *d* = 0.96; anxiety: *p* < 0.001, *d* = 0.91). Regarding burnout symptomatology, emotional exhaustion was significantly reduced for women (*p* = 0.004, *d* = 0.68). Personal fulfillment at work also increased after training (*p* = 0.013, *d* = 0.56) while cynicism was not significantly changed (*p* = 0.15, *d* = 0.32); however, the improvements showed a small effect size. For men, decreased mean scores on the emotional exhaustion scale were observed after the intervention (*p* = 0.04, *d* = 0.43). Symptom distress also decreased but not significantly (PHQ-9: *p* = 0.20, *d* = 0.26; GAD-7: *p* = 0.32, *d* = 0.20; PHQ-15: *p* = 0.67, *d* = 0.09).

## 4. Discussion

To the best of our knowledge, this study is one of few evaluations of an intensive work-related group therapy for work-related mental disorders.

### 4.1. Reduction in Symptoms of Work-Related Mental Disorders and Burnout Dimensions

The main result of our study is the significant and clinically relevant reduction in symptoms of work-related mental disorders such as depression and anxiety. This is consistent with our assumptions and with psychotherapy research [28,29]. Many of the methods used were specifically aimed at reducing depressive symptoms (working out basic needs, psychoeducation, coping with stressful thoughts). The group setting also offered the opportunity to discuss and try out anxiety-provoking situations, which were then implemented in everyday life. This could also explain the symptom reduction. Somatization was not significantly reduced in the total sample. The literature shows that psychotherapeutic interventions are helpful in treating medically unexplained symptoms [30]. In our study, however, it is not entirely clear whether the complaints reported are fully explained by somatization. Due to the rather older age composition of the sample (50% being over 50 years old), physical symptoms could also be due to chronic age-related changes. In addition, major parts of the contents of the intervention were depression- or anxiety-specific but less related to physical symptoms. After the intervention, emotional exhaustion was decreased, while the other burnout dimensions did not change significantly. Cynicism only tended to be reduced in the sample, and personal fulfillment could hardly be increased. This could be explained by the special sample. Cynicism and low personal fulfillment are mainly associated with occupations that offer few resources according to the Job Demands–Resources Model [31,32]. Cynicism is the burnout dimension most likely to be associated with leaving a job [33]. Erlangen is the city with the third highest median income in Germany [34]. Due to the large industrial, research and educational companies as well as medical institutions in the study location, the sample consisted of employees with (highly) qualified jobs. These positions could act as a protective factor against cynicism due to the increased opportunities for job resources. On the other hand, cynicism is described as decreasing compassion and empathy with clients [24]. This could particularly affect members of social professions who rely on a certain level of empathy to do their job. Since our study sample predominantly consisted of people doing office work, cynicism might be less crucial as a burnout measure. The same could be the reason for the limited change in personal fulfillment. In addition, the majority of the participants worked in jobs in leading positions, which are usually associated with more job control. This is consistent with the fact that subjective personal fulfillment was already high in the sample. Future research should address the question whether cynicism and personal fulfillment can be improved after the intervention described here in a more diverse sample.

### 4.2. Gender Differences

For women, the scores in somatization, depression and anxiety decreased substantially, showing large effect sizes (*d* = 0.76–0.91), but also emotional exhaustion was significantly and clinically relevant reduced and personal fulfillment increased. In contrast, men only benefited in terms of emotional exhaustion. Although this is an important finding in the attempt to improve workplace-related symptomatology, the question remains why women seem to have benefited significantly more from the training. Gender differences in outcomes from psychotherapeutic care have long been documented [35]. Women are more likely to benefit from therapeutic treatments than men [36]. Because of their socialization and tendency to share emotional information as described in the model of “tend and befriend”, women could benefit more from group therapy. If women are more inclined to share in the group and to apply insights in their daily lives both within and between sessions, this could play a role in explaining more improvements among women than men. Furthermore, in consistency with the literature, the women generally showed higher levels of distress and therefore had more potential for improvement.

### 4.3. Strengths and Limitations

To our knowledge, this is the first naturalistic study to evaluate a group therapy regarding mental health symptoms and work-related symptoms. All participants received a comprehensive intervention of 12 sessions. They were diagnosed and treated according to a standardized procedure by qualified experts. Nevertheless, our study has some limitations. It was a naturalistic study without a control group. This was decided for pragmatic reasons and to improve ecological validity. Future studies should include a control group to verify the superiority of the intervention over care as usual. The aim of the intervention was to strengthen employees’ personal resources so that they can better cope with work-related stress or disorders. Thus, the study is limited by focusing only on individual employees and what they can do to cope with stress and does not address root causes of work-related stress (work factors in interplay with personal factors). This limitation applies to individual psychotherapy in general, but this could be improved through cooperation with companies or interdisciplinary exchange with occupational physicians. A more in-depth analysis of the mechanisms of employees’ work behavior that influences mental health could provide further implications. Due to the small sample and the lack of data on the individual work mechanisms of the group participants, this could not be examined in this study. Future research should address this question through qualitative research. To answer these questions, an in-depth analysis of the work mechanisms that lead to mental illness and the possibilities for influencing the workplace versus only the individual is currently being conducted in a randomized controlled trial [14]. Due to the small sample size, a generalization of the results is not possible. The power is also limited to detecting small effects, in particular in the subsamples of women and men. Unfortunately, a larger sample size was not possible for pragmatic reasons, as this was an extensive intervention with limited resources. At the same time, the survey period overlapped with the COVID-19 pandemic, which made regular group therapy difficult.

## 5. Conclusions

The results of the present study showed reduced symptom severity of work-related mental disorders such as depression and anxiety as well as emotional exhaustion after a work-related psychotherapeutic group intervention. Our results indicate that women, who are more frequently and more severely affected by mental distress, can be supported by such an intervention. This has concrete implications for the health sector. The offer of concrete work-related psychotherapeutic treatment should be expanded in clinics and psychotherapeutic practices. The focus on the workplace should increasingly be included in inpatient psychotherapy so that the relevant work-related factors that contribute to mental illness can be identified and treated. In addition, there should be more interdisciplinary exchange between therapists and occupational health practitioners to improve the situation of patients at their workplace. Future studies should add a control group to the design in order to draw causal conclusions. Furthermore, the needs of men in the context of work-related mental disorders should possibly be elicited through qualitative studies so that an adapted intervention can be offered. Future studies may experiment with different types of group compositions to better understand the circumstances under which men may achieve the best outcomes from such therapy (e.g., all male groups, gender-balanced groups). More research is also required on work-related mental disorders and their differentiation from other mental illnesses and possible cost-effective implementation in the health care system.

## Figures and Tables

**Figure 1 ijerph-20-02673-f001:**
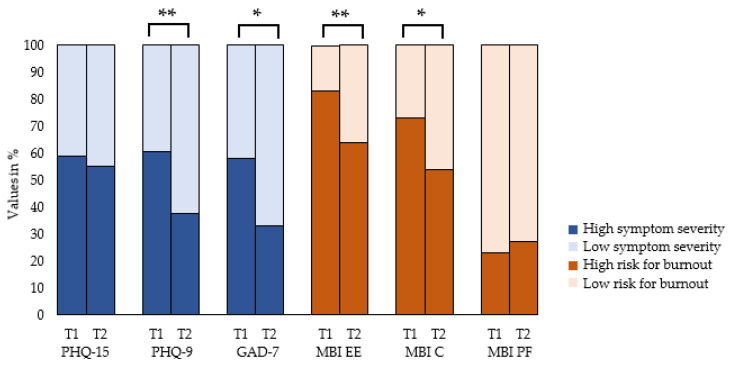
Symptom change from T1 to T2. PHQ-15 (Patient Health Questionnaire Somatization Module); PHQ-9 (Patient Health Questionnaire Depression Module); GAD-7 (General Anxiety Disorder Short Scale); MBI (Maslach Burnout Inventory); EE (emotional exhaustion); C (cynicism), PF (personal fulfillment); * *p* < 0.05; ** *p* < 0.01.

**Table 1 ijerph-20-02673-t001:** Sociodemographic and occupational characteristics of the study sample.

	Total Sample *N* = 48
**Gender, *n* (%)**	
Women	23 (47.9)
Men	25 (52.1)
**Age, years, *n* (%)**	
26–40	8 (16.7)
41–50	16 (33.3)
51–60	24 (50.0)
**Marital status, *n* (%)**	
Single	11 (22.9)
Married	23 (47.9)
Divorced	11 (22.9)
Separated	2 (4.2)
Missing	1 (2.1)
**Children, *n* (%)**	
Yes	30 (62.5)
No	18 (37.5)
**Migration background, *n* (%)**	
Yes	5 (10.4)
No	43 (89.6)
**Education**	
Primary School	3 (6.3)
Middle School	19 (39.6)
Secondary school	26 (54.2)
**Professional qualification**	
Apprenticeship	18 (37.5)
Master (craftsmen)	7 (14.6)
University	19 (39.6)
None	1 (2.1)
Other	3 (6.3)
**Professional group**	
Skilled worker	2 (4.2)
Employee with basic tasks	2 (4.2)
Employee with qualified tasks	24 (50.0)
Employee with high qualified tasks/ leadership	14 (29.2)
Employee with extensive management and decision-making authority	5 (10.4)
Other	1 (2.1)
**Employment status**	
Full-time	36 (75.0)
Part-time	12 (25.0)

**Table 2 ijerph-20-02673-t002:** Number of ICD-10 coded mental and behavioral disorders of the total sample.

	Total Sample *N* = 48
**Number of mental and behavioral disorders, *n* (%)**	
One	32 (66.7)
Two	12 (25.0)
Three	4 (8.3)
**Main ICD-10 diagnoses, *n* (%)**	
**Mood (affective) disorders (F3)**	30 (62.5)
Mild depressive episode (F32.0)	5 (10.4)
Moderate depressive episode (F32.1)	10 (20.8)
Recurrent depressive disorder, current episode mild (F33.0)	4 (8.3)
Recurrent depressive disorder, current episode moderate (F33.1)	10 (20.8)
Recurrent depressive disorder, current episode severe without psychotic symptoms (F33.2)	1 (2.1)
**Neurotic, stress-related and somatoform disorders (F4)**	13 (56.3)
Panic disorder (F41.0)	1 (2.1)
Generalized anxiety disorder (F41.1)	1 (2.1)
Mixed anxiety and depressive disorder (F41.2)	1 (2.1)
Adjustment disorders (F43.2)	5 (10.4)
Other reactions to severe stress (F43.8)	2 (4.2)
Undifferentiated somatoform disorder (F45.1)	2 (4.2)
Somatoform autonomic dysfunction (F45.3)	1 (2.1)
**Behavioral syndromes associated with physiological disturbances and physical factors (F5)**	3 (6.3)
Eating disorder, unspecified (F50.9)	1 (2.1)
Psychological and behavioral factors associated with disorders or diseases classified elsewhere (F54)	2 (4.2)
**Disorders of adult personality and behavior (F6)**	2 (4.2)
Other specified disorders of adult personality and behavior (F68.8)	2 (4.2)
**Total number of ICD-10 diagnoses, *n* (%)**	
Mood (affective) disorders (F3)	39 (60.9)
Neurotic, stress-related and somatoform disorders (F4)	17 (26.5)
Behavioral syndromes associated with physiological disturbances and physical factors (F5)	4 (6.3)
Disorders of adult personality and behavior (F6)	4 (6.3)

**Table 3 ijerph-20-02673-t003:** Symptoms at admission and discharge.

T1 **							T2 **					
	Total Sample (*n* = 48) M (SD)	Women (*n* = 23) M (SD)	Men (*n* = 25) M (SD)	Comparison between Men and Women at T1			Total Sample (*n* = 48) M (SD)	Women (*n* = 23) M (SD)	Men (*n* = 25) M (SD	Comparison between Men and Women at T2		
				t	p	d				t	p	d
**PHQ-15**	10.81 (5.61)	12.74 (5.43)	9.04 (5.27)	**−3.99**	**<0.001**	**−0.95**	9.85 (6.19)	10.30 (5.80)	9.44 (6.62)	**−2.00**	**<0.05**	**−0.48**
**PHQ-15 ***	10.73 (5.56)	12.57 (5.38)	9.04 (5.27)	**−3.68**	**<0.001**	**−0.88**	9.73 (6.12)	10.04 (5.67)	9.44 (6.62)	−1.7	0.09	−0.41
**PHQ-9**	11.06 (6.19)	12.25 (5.97)	9.96 (5.50)	−1.20	0.23	−0.29	8.29 (6.27)	8.17 (5.75)	8.40 (6.83)	0.28	0.78	0.07
**GAD-7**	9.94 (5.18)	10.74 (4.85)	9.20 (5.45)	−1.65	0.10	−0.39	7.13 (5.69)	6.17 (4.50)	8.00 (6.57)	0.33	0.74	0.08
**MBI EE**	4.63 (0.95)	4.87 (0.77)	4.42 (1.07)	**−2.31**	**<0.05**	**−0.55**	4.05 (1.35)	4.14 (1.16)	3.97 (1.52)	−0.89	0.38	−0.21
**MBI Z**	3.93 (0.99)	4.00 (0.86)	3.86 (1.11)	−0.17	0.87	−0.04	3.70 (1.32)	3.70 (1.19)	3.72 (1.44)	1.11	0.28	0.27
**MBI PF**	4.30 (0.83)	4.31 (0.74)	4.27 (0.93)	0.69	0.49	0.16	4.41 (0.94)	4.62 (0.71)	4.23 (1.09)	−0.66	0.51	−0.16

PHQ-15 (Patient Health Questionnaire Somatization Module); * without Menstruation Item; PHQ-9 (Patient Health Questionnaire Depression Module); GAD-7 (General Anxiety Disorder Short Scale); value of completer sample. ** Only the completer sample is represented. Varying sample sizes; significant *p*-values are marked in bold.

**Table 4 ijerph-20-02673-t004:** Symptom change of the total completer sample, women, men.

	Total Sample T1–T2			Women T1–T2			Men T1–T2		
	t	p	d	t	p	d	t	p	d
**PHQ-15**	1.57	0.12	0.23	3.65	**<0.001**	**0.76**	−0.43	0.67	−0.09
**PHQ-15 ***	1.61	0.11	0.23	3.59	**<0.05**	**0.75**	−0.43	0.67	−0.09
**PHQ-9**	3.6	**<0.001**	**0.53**	4.62	**<0.001**	**0.96**	1.32	0.20	0.26
**GAD-7**	3.4	**0.001**	**0.49**	4.36	**<0.001**	**0.91**	1.02	0.32	0.20
**MBI EE**	3.8	**<0.001**	**0.55**	3.26	**<0.05**	**0.68**	2.13	**<0.05**	**0.43**
**MBI C**	1.50	0.14	0.22	1.51	0.15	0.32	0.65	0.52	0.13
**MBI PF**	−1.20	0.24	−0.17	−2.70	**<0.05**	**−0.56**	0.41	0.68	0.08

PHQ-15 (Patient Health Questionnaire Somatization Module); * without Menstruation Item; PHQ-9 (Patient Health Questionnaire Depression Module); GAD-7 (General Anxiety Disorder Short Scale); MBI EE, C, PF (Maslach Burnout Inventory Emotional Exhaustion, Cynicism, Personal Fulfillment); significant *p*-values and the corresponding effect sizes are marked in bold.

## Data Availability

The data presented in this study are available on request from the corresponding author.

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
