# Peer review of "Evaluation of a Group Therapy for Work-Related Mental Disorders"

_ijerph, 2023, doi:10.3390/ijerph20032673_

Round 1
Reviewer 1 Report
Review of ‘Evaluation of a group therapy…’ by Hondong et al.
General:
Well written report of a small-sample straightforward study of mental health and burnout before and after a group intervention for employees. While I lean towards recommending the manuscript for publication, the authors should avoid the use of cause-effect language (currently a major problem), given that the study design does not support such conclusions to be made. Stating e.g., that the intervention reduced symptoms is problematic, while stating that symptoms were reduced from before to after the treatment is OK.
Title:
Reconsider the use of expressions such as ‘occupational mental health problems’, and similar (e.g., work-related mental disorders) expressions throughout the manuscript. Medical diagnoses are purely descriptive, but expressions such as the ones mentioned at least add connotations about what causes the relevant problem (work factors). I suggest the authors treat these and similar expressions with much caution.
Abstract:
Both the PHQ-9 and PHQ-15 are used in the study. While I assume they are not conceptually overlapping, the differences between the measures should be described as part of section 2.4.4.
Line 19: 48 participants completed (remove have, write consistently in past tense)
Line 25-26: remove the sentence about cut-off values; this should only be described in the Methods/Measures section (too detailed for the abstract).
Line 26-27: Include the comparison (women benefited more than men), and results for men to substantiate the statement.
Line 28: The first sentence in the conclusion section describes cause and effect, which is inappropriate given the study design (no control group). Instead, state that the participants’ mental health symptoms were substantially reduced during the course of the work-related group therapy (which is similar, but without overstating the importance of the treatment)
Introduction:
Lie 44-45: Related to the first comment (title). Reconsider the use of the expression ‘occupational mental disorders’ (the importance of the work setting for mental disorders?). See if you can find expressions that are more neutral with regard to the problem cause.
Line 49: ‘Here’ – what does it refer to? Increase precision, e.g., ‘in that context’.
Line 55: what does it mean to ‘improve workplace-related factors’? Increase precision, be specific.
Line 75: what gender differences have been shown? Be more specific.
Methods:
Participants:
Lines 83-87: In these sentences, the authors define what they mean by the expression ‘occupational mental disorder’. As previously noted, I believe the expression should be changed to avoid cause-effect connotations. However, also the definition (‘common mental disorders with an explicit reference to workplace’) should be clarified. What does this mean, in specific terms? In addition, clarify whether inclusion required all three criteria to be fulfilled (CMD + Z56 + Z73), or just two of them (CMD + [Z56 or Z73]), and whether all participants were currently employed (i.e., none on sick leave/benefits?).
Section 2.4.1: The ethics section is misplaced under ‘Measures’.
Line 133: ‘were’ assessed. Please use past tense consistently when describing things that took place.
Section 2.4.3. With reference to Table 1 information about ‘Professional group’, were these the response options used in the questionnaire? How would a participant know which category he or she belonged to? The opposite of a simple task is not a ‘qualified’ task, but a ‘difficult’ one. What is the difference between high qualified leadership and extensive leading position? The response options are blurry; please clarify both in section 2.4.3 and in Table 1.
Section 2.4.4: Provide reference to the last sentence about cut-off values.
Section 2.4.5: Provide reference to the last sentence about cut-off values.
Section 2.5 Analysis: line 165, ‘relative frequencies’, do you mean percentages?
Line 166: you do not ‘indicate’ data, rephrase into ‘were calculated to describe the sample’
Results:
Section 3.1, line 181-182: the sample of completers
Section 3.2, line 186-187: almost half were married (47.9%), while smaller proportions were single (22.9%) or divorced (22.9%)
Table 2: Two separate sections of the table have the same heading (‘Number of ICD-10 coded mental and behavioral disorders’). I understand the first section, which actually reports on how many participants had how many diagnoses, but not the second one. Please clarify – either way, use headings that align with the relevant content. Related to the next section about main diagnoses, I urge the authors to report the specific diagnoses and not the codes for these diagnoses. Summaries for each section (each diagnostic group) should come after the specific diagnoses, not before.
Section 3.4: line 207: M and SD for PHQ-9 do not correspond with figures in Table 3.
Table 3: t, not T. Clarify that statistical tests are comparisons between men and women at each time point, and not comparisons across time. In the table note, remove information about PCL-5 (which is not included in the study).
Section 3.6, line 229: Symptoms of ‘occupational distress’ – given the MBI results being reported, I believe you should maintain the terminology and stick with ‘burnout’ instead of introducing new concepts.
Figure 1: Consider reversing the scores on MBI-PF so that it better aligns with the other bars in the diagram. Add ‘PF’ to the relevant label and indicate in the explanatory note the meaning of * and **.
Table 4:
Heading: The table is not split into younger/older, as far as I can see.
Table 4: remove underscore (beneath t, p, d, and again t) in columns 2-5.
Discussion:
Line 272-273: If women are more inclined to share in the group, and to apply insights in their daily lives both within and between sessions, this could play a part in explaining more improvements among women than men. Consider adding to section 4.2, where you discuss along similar lines.
Line 278: older age composition of the sample
Line 279-280: …, major parts of the contents of the intervention were …
Line 284: low personal fulfilment
Line 287: What does ‘due to the industrial structures of the study location’ mean? Clarify the meaning
Line 291: members of social professions (add reference)
Line 294-295: This is consistent with the fact…
Line 296: Specify which question should be addressed.
Section 4.2, line 299-300: …, but also emotional exhaustion was significantly….
Line 304: Gender differences in outcomes from psychotherapeutic care…
Line 305: Women are more likely to benefit from…
Study limitations: Authors state that the aim of the intervention was to strengthen employees’ personal resources so that they can better cope with work-related stress or disorders’. Thus, the study is limited by focusing only on individual employees and what they can do to cope with stress and does not address root causes of work-related stress (work factors in interplay with personal factors). This should be included as a limitation.
line 315-317: Rewrite your argument. Stating 1) ‘control group was omitted due to ethical reasons’ is in direct contrast to stating 2) ‘future studies should include a control group’. Obviously, statement 1) needs to be reworked. Why did you not include a control group? Pragmatic reasons (time, resources etc.) are fine, as long as you maintain a logical, cohesive argument.
The authors state that the response rate (completers) could introduce selection bias. I would encourage them to examine this formally (instead of just opening up the possibility for it) by testing relevant differences between completers and non-completers.
Line 322: Due to the small sample, …
Conclusion section: avoid causal language (‘effectiveness’, etc). Future studies may experiment with different types of group compositions to better understand the circumstances under which men may achieve the best outcomes from such therapy? All male groups, gender-balanced groups, other options? Just a suggestion.
References: In-text references are formatted (Author, Year), while references in the reference list are formatted by number. This needs to be changed according to journal requirements.
Supplement 3 and 4: remove information about PCL-5
Supplement 4: indicate whether top/bottom parts of the table relate to younger/older participants.
Author Response
Thank you very much for your review. You will find our detailed response in the attached document.

Reviewer 2 Report
1. Why is occupational mental health selected in this manuscript from the perspective of a German employee?
2. What are the main scientific questions addressed by this research?
3. A more in-depth analysis of the mechanism of employee work behavior that influences mental health is needed to analyze the results and implications of this study's theoretical value.
4. A more in-depth and targeted analysis of practical implications is needed to provide a better reference or guide for the case area.
5. Some limitations in the analysis of the study area, the logical theoretical relationship of mental health and employee behavior, the basis of the plot, and a selected sample of 48 should be clarified, for research with statistics, the ideal sample size should be at least 100, and the targeted suggestions should be more clearly illustrated.
Author Response

(The authors gave the same response as above.)

Round 2
Reviewer 2 Report
The author has not explained regarding determining the location of the research, I know the research was conducted in Germany, but Germany is wide; moreover, the sample is very lacking for quantitative studies. So it is necessary to explain the reasons for the detailed research location. In addition, with a limited number of samples, it is advisable to conduct a qualitative study because these findings cannot be generalized for quantitative studies.
Author Response
Thank you for the comments. You can find our detailed answer in the attached word document.
